# Nomograms of human hippocampal volume shifted by polygenic scores

Mohammed Janahi[1,2]*, Leon Aksman[3], Jonathan M Schott[4], Younes Mokrab[2,5], Andre Altmann[1], On behalf of for the Alzheimer's Disease Neuroimaging Initiative

[1]Centre for Medical Image Computing (CMIC), Department of Medical Physics and Biomedical Engineering, University College London, London, United Kingdom; [2]Medical and Population Genomics Lab, Human Genetics Department, Research Branch, Sidra Medicine, Doha, Qatar; [3]Stevens Neuroimaging and Informatics Institute, Keck School of Medicine, University of Southern California, Los Angeles, United States; [4]Dementia Research Centre (DRC), Queen Square Institute of Neurology, University College London, London, United Kingdom; [5]Department of Genetic Medicine, Weill Cornell Medicine-Qatar, Doha, Qatar

*For correspondence:
Rmapmja@ucl.ac.uk

Competing interest: The authors declare that no competing interests exist.

**Abstract** Nomograms are important clinical tools applied widely in both developing and aging populations. They are generally constructed as normative models identifying cases as outliers to a distribution of healthy controls. Currently used normative models do not account for genetic heterogeneity. Hippocampal volume (HV) is a key endophenotype for many brain disorders. Here, we examine the impact of genetic adjustment on HV nomograms and the translational ability to detect dementia patients. Using imaging data from 35,686 healthy subjects aged 44–82 from the UK Biobank (UKB), we built HV nomograms using Gaussian process regression (GPR), which – compared to a previous method – extended the application age by 20 years, including dementia critical age ranges. Using HV polygenic scores (HV-PGS), we built genetically adjusted nomograms from participants stratified into the top and bottom 30% of HV-PGS. This shifted the nomograms in the expected directions by ~100 mm$^3$ (2.3% of the average HV), which equates to 3 years of normal aging for a person aged ~65. Clinical impact of genetically adjusted nomograms was investigated by comparing 818 subjects from the Alzheimer's Disease Neuroimaging Initiative (ADNI) database diagnosed as either cognitively normal (CN), having mild cognitive impairment (MCI) or Alzheimer's disease (AD) patients. While no significant change in the survival analysis was found for MCI-to-AD conversion, an average of 68% relative decrease was found in intra-diagnostic-group variance, highlighting the importance of genetic adjustment in untangling phenotypic heterogeneity.

## Editor's evaluation

This manuscript considers whether genetic information can improve the clinical utility of population norms derived from brain imaging data. The authors propose to incorporate polygenic scores into normative models of hippocampal volume to improve predictions of neurodegenerative disease. This approach is elegantly demonstrated in this manuscript and may be useful for clinical translation of population neuroimaging.

## Introduction

Brain imaging genetics is a rapidly evolving area of neuroscience combining imaging, genetic, and clinical data to gain insight into normal and diseased brain morphology and function (*Shen and Thompson, 2020*). Normative modelling is an emerging method in neuroscience, aiming to identify

cases as outliers to a distribution of healthy controls and was shown to have potential to improve early diagnosis, progression models, and risk assessment (*Marquand et al., 2016*; *Pinaya et al., 2020*; *Wolfers et al., 2020*; *Ziegler et al., 2014*). Where conventional case-control studies generally require both cases and controls to be well clustered, normative models work well even when cases are not clustered or overlap with controls. Nomograms are a common implementation of normative models and have been used as growth charts of brain volumes across age in both developing and aging populations (*Castellanos et al., 2002*; *Scahill et al., 2003*; *Peterson et al., 2018*).

Normative modelling identifies cases by their deviation from normality, however, genetics shapes what is 'normal'. Heritability studies have found that whole brain volume is 90 ± 4.8 heritable (*Lukies et al., 2017*), hippocampal volume (HV) is 75 ± 5 (*Kremen et al., 2010*; *Thompson et al., 2020*; *Hibar et al., 2015*), and other cortical brain areas between 34% and 80% (*Rentería et al., 2014*; *Zhao et al., 2019*). Genome-wide association studies (GWASs) have identified genome-wide significant variants that explain 13 ± 1.5 of the variation in HV (*Hibar et al., 2017*), 34 ± 3 in total cortical surface area, and 26±2% in average cortical thickness (*Grasby et al., 2020*). The gap between estimates from GWAS hits and formal heritability estimates (termed the 'missing heritability') (*Manolio et al., 2009*) implies that less significant variants also have an influence and that it may be captured through polygenic scores (PGSs) (*Foo et al., 2021*; *Axelrud et al., 2018*; *Escott-Price et al., 2015*). In this work we demonstrate the impact of accounting for polygenic effects in normative modelling of HV.

Damage to the hippocampus which is integral to memory processes (*Bird and Burgess, 2008*) has been associated with major depressive disorder (*Bremner et al., 2000*), schizophrenia (*Nelson et al., 1998*), epilepsy (*Whelan et al., 2018*), and Alzheimer's disease (AD) (*Pini et al., 2016*). AD is a global health burden: 7% of people over 60 are diagnosed with dementia (*van der Flier and Scheltens, 2005*) of which AD accounts for 70% (*Rabinovici, 2019*). The pathophysiological processes underlying AD, namely amyloid and tau pathology accumulation, are thought to precede brain atrophy, which typically starts in the hippocampus and medial temporal lobe and then spreads throughout the neocortex (*Rabinovici, 2019*).

The normal variation of HV is of great clinical interest as the early and often prominent hippocampal atrophy seen in AD creates a need for early diagnosis and disease tracking. Many studies have examined HV across age (*Schmidt et al., 2018*; *Fraser et al., 2015*), for example, a recent study by *Nobis et al., 2019*, produced HV nomograms from UK biobank (UKB) for use in clinical settings. It is

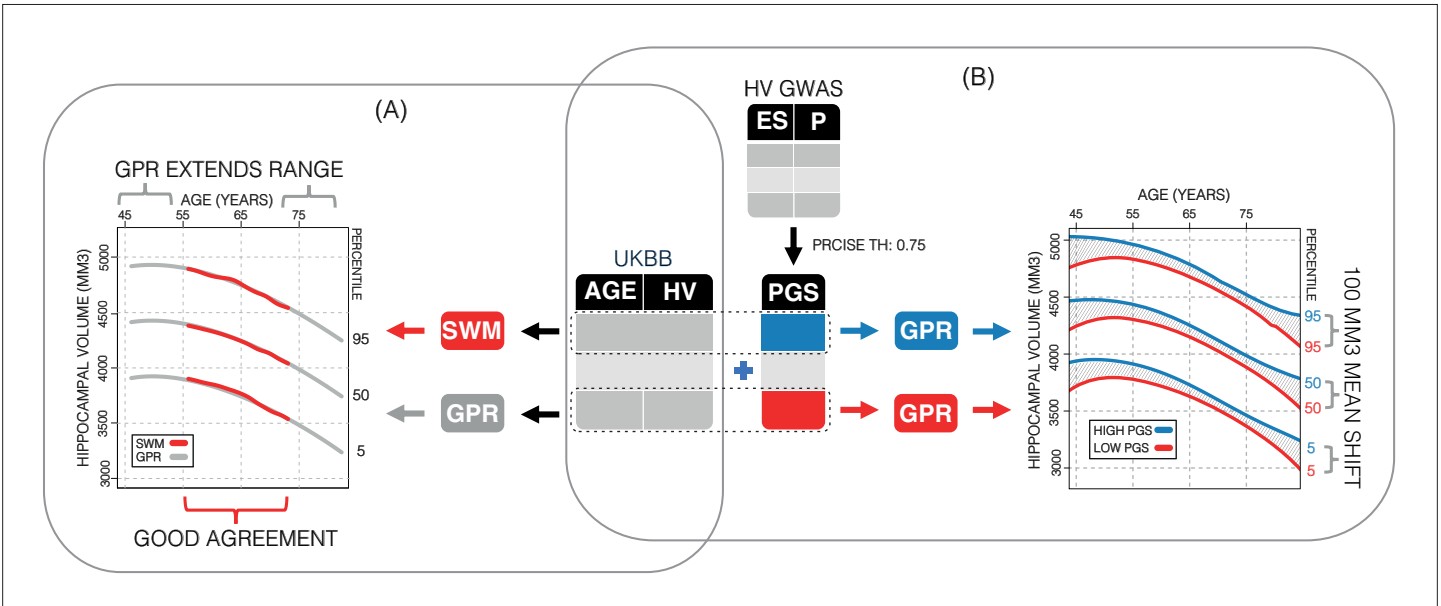

**Figure 1.** Study overview. (**A**) Using 35,686 subjects from the UK Biobank, we generate nomograms using two methods: a previously reported sliding window method (SWM) and Gaussian process regression (GPR). We find that GPR is more data efficient than the SWM and can extend the nomogram into dementia critical age ranges. (**B**) Using a previously reported genome-wide association study, we generate polygenic scores (PGSs) for the subjects in our UK Biobank table. We then stratify the table by PGS and generate nomograms for the top and bottom 30% of samples separately. We find the genetic adjustment differentiates the nomograms by an average of 100 mm³, which is equivalent to about 3 years of normal aging for a 65-year-old.

important to note that some of the variation in the normative models can be explained by the clear impact of genetics on HV (*Hibar et al., 2017*; *Mather et al., 2015*). Thus far, the few attempts at including genetics in the construction of HV nomograms have focussed on disease-related variants. For instance, two recent studies examined the impact of the AD-associated *APOE* gene (*Ching et al., 2020*; *Veldsman et al., 2021*), showing that APOE4/4 carriers had significantly lower HV trajectories. This effect is likely driven by AD-related disease processes since APOE4/4 carriers have a 10-fold risk of developing AD (*Kim et al., 2009*; *Liu et al., 2013*). However, the genetic impact on variation in HV in healthy population remains underexamined in the context of nomograms. In this work, we close this gap. We built HV nomograms using a GPR method (*Figure 1A*). We then computed a PGS of HV for subjects in our cohort and built genetically adjusted nomograms (*Figure 1B*). We found that genetic adjustment did in fact shift the nomograms and that, because the model requires no smoothing, our GPR nomograms provided an extended age range compared to previous methods.

## Results

In the UKB sample, 453 subjects were excluded for various conditions, 3497 for genetic ancestry, and 28 subjects were outliers: leaving a total of 35,686 subjects. In the Alzheimer's Disease Neuroimaging Initiative (ADNI) application dataset, 26 subjects were excluded for genetic ancestry, and 314 based on HV quality scores: leaving 818 subjects.

### SWA vs. GPR for nomogram estimation

Nomograms of healthy subjects generated using the sliding window approach (SWA) and GPR method displayed similar trends (*Figure 2*; *Figure 2—figure supplement 2*). However, GPR nomograms spanned the entire training dataset age range (45–82 years) compared to the SWA (52–72 years). This is primarily because the SWA is a non-model-based approach that requires smoothing to avoid edge effects, and a Gaussian smoothing window of width 20 was used (*Nobis et al., 2019*). This extension allowed 86% of all diagnostic groups from the ADNI to be evaluated vs. 56% in the SWA nomograms (*Figure 2*; *Figure 2—figure supplement 2*). Furthermore, our GPR nomograms confirmed previously reported trends: Overall, the average 50th percentile in male nomograms (4162 ± 222) was higher than the female nomograms (3883 ± 170), and within each sex, right HV was larger than left HV (*Figure 2*; *Figure 2—figure supplement 2*). We also observed that along the 50th percentile, male HV declined faster ($-20.3 mm^3$/year) than female HV ($-14.6 mm^3$/year). Additionally, in GPR nomograms, HV peaks in women at age 53.5 years with a less pronounced peak in males at 50 years (*Figure 2*; *Figure 2—figure supplement 2*). Training the GPR model with 16,000 samples took ~1 hr on a consumer grade machine (2.3 GHz 8-Core Intel Core i9).

### PGS for HV

The calculated PGS, based on an earlier GWAS for average bilateral HV (*Hibar et al., 2017*), as expected, showed a strong correlation with HV in the UKB data. Overall, the PGSs showed a significant positive correlation with HV across all p-value thresholds and training sample subsets ($p < 2.7E-24$; *Table 1*). PGSs explained more variance in males vs. females. Furthermore, PGSs did not show detectable differences in left vs. right HV; and explained the most variance in mean bilateral HV (*Table 1*, *Figure 3—source data 1*). In all tested settings, the explained variance ($R^2$) by the PGS across p-value threshold was similar: with one peak at the 1E-7 threshold (capturing few but very significant SNPs), a higher peak at the 0.75 threshold (capturing many SNPs with mostly small effect sizes) (*Figure 3*). For the ADNI dataset, this distribution increased with the threshold. When investigating mean HV across percentile of PGS at the 0.75 threshold (highest $R^2$), the top and bottom 20% of scores accounted for 41% of the variance in HV (*Figure 3*) with similar values observed across thresholds in both datasets (*Figure 3—figure supplements 1 and 2*).

### Genetics stratified nomograms

We will focus on the p-value threshold of 0.75 as it achieved best or close to best performance overall (*Figure 3—source data 1*). Genetics had a clear effect on the nomograms: the high PGS nomograms were shifted upwards while the low-PGS nomograms were shifted downwards; an effect which could be observed at both the model and data level (*Figure 4*; *Figure 4—figure supplement 3*), both by

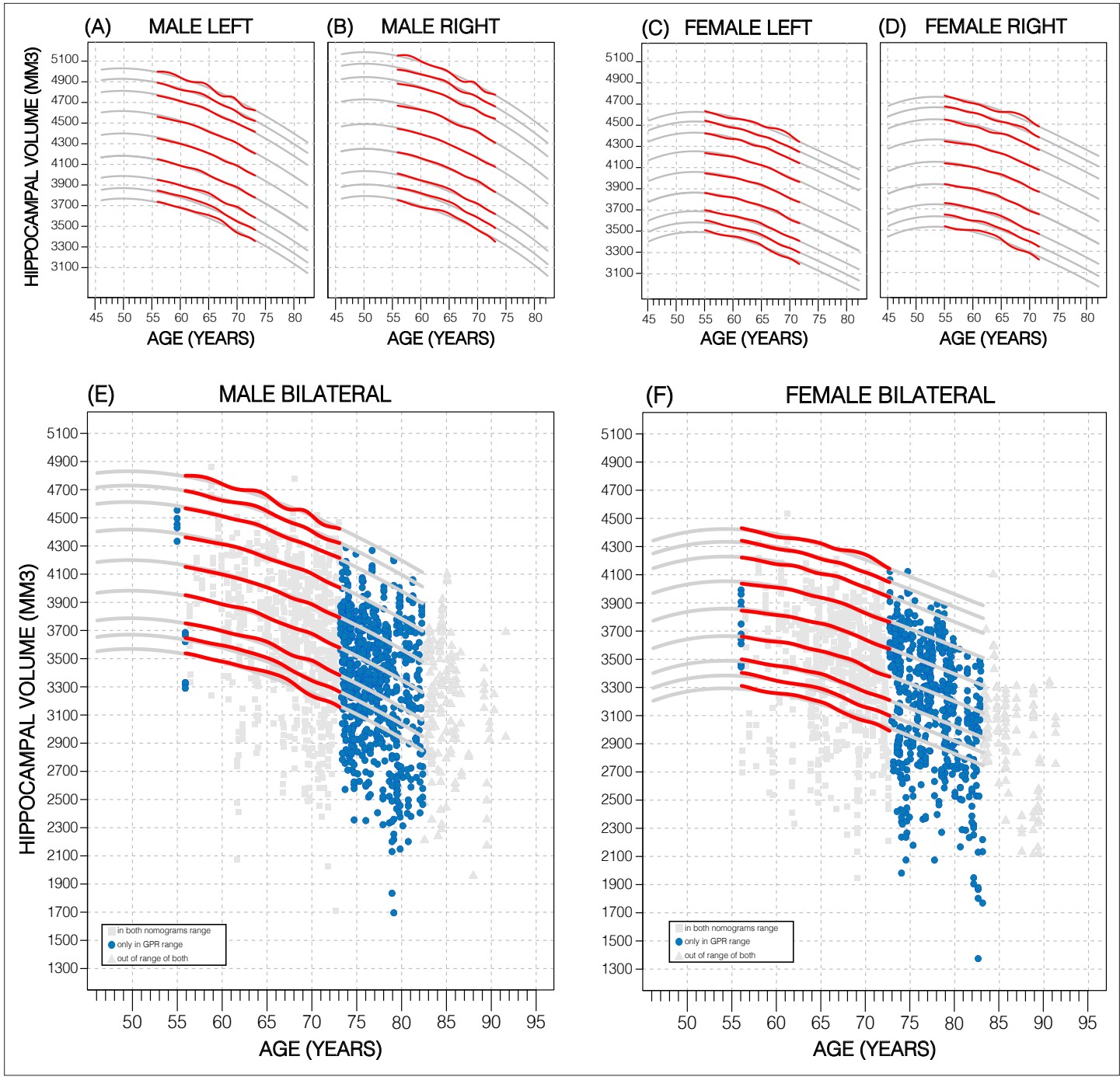

**Figure 2.** Comparing nomogram generation methods. Nomograms produced from healthy UK Biobank (UKB) subjects using the sliding window approach (SWA) (red lines) and Gaussian process regression (GPR) method (grey lines) show similar trends. Both left hemisphere nomograms (**A, C**) are lower than their right counterparts (**B, D**). Male nomograms are higher than female nomograms (A vs. C) and (B vs. D). Female hippocampal volume (HV) shows a peak at 53.5 years of age, while male HV shows a less prominent peak at 50 years of age. SWA and GPR show good agreement, while GPR enables a 10-year nomogram extension in either direction. The benefits of this extension can be seen with scatter plots of Alzheimer's Disease Neuroimaging Initiative (ADNI) subjects of all diagnoses overlayed (**E, F**). The extended age range of the GPR nomograms (45–82 years) enables the evaluation of an additional 43% of male data (**E**) and 34% of female data (**F**) (turquoise circles). A similar figure with only the cognitively normal ADNI subjects can be found in *Figure 2—figure supplement 2*.

The online version of this article includes the following figure supplement(s) for figure 2:

**Figure supplement 1.** Expanded Gaussian process regression (GPR) nomogram.

**Figure supplement 2.** Model fit of healthy Alzheimer's Disease Neuroimaging Initiative (ADNI) subjects.

**Figure supplement 3.** Performance of Gaussian process regression (GPR) and sliding window method (SWM) across sample size.

**Figure supplement 4.** Gaussian process regression (GPR) model across top/bottom thresholds.

around 1.2% of the average HV (50 mm$^3$). Thus, the difference between high and low PGS nomograms was ~2.3% of the average HV (100 mm$^3$). An ANOVA test of the percentiles produced with the adjusted vs. unadjusted nomograms revealed that the groups were significantly different to each other ($F$>19; p<8.04E-6; *Table 2*). The HV peak previously observed at 50 years in males was less pronounced in the high PGS nomogram and more so in the low PGS nomogram (*Figure 4*, *Figure 4—figure supplement 1*). Adjusting nomograms using ICV and AD PGSs, instead of HV PGS, did not result in nomograms that were meaningfully different from the non-adjusted nomograms (*Figure 4—figure supplement 2*).

## External evaluation on ADNI data

In the ADNI dataset we investigated whether the shift in genetically adjusted nomograms could be leveraged for improved diagnosis. Using the non-adjusted nomogram, cognitively normal (CN) participants (*n*=225) had a median bilateral HV percentile of 61% (±25% SD), mild cognitive impairment (MCI) participants (*n*=391) had 25% (±26% SD), and AD participants (*n*=121) had 1% (±9% SD) (*Figure 5*). Visual inspection revealed that while CN participants were spread across the quantiles, AD participants lay mostly below the 2.5% quantile, and MCI participants spanned the range of both CN and AD participants (*Figure 4*). Bisecting the samples by PGS showed that high PGS CN samples had median percentiles of 65% (±27% SD) and low PGS had 54% (±26% SD). When comparing the same samples against the genetically adjusted nomograms instead, high PGS CN samples had 60% (±26% SD) and low PGS had 59% (±26% SD). Thus, reducing the gap between high and low PGS CN participants by 9% (from 10% to 1%, a 90% relative reduction). Similar analysis showed a reduction in MCI participants by 10% (60% relative reduction), and 0.5% (56% relative reduction) in AD participants. The above effects persisted across most strata (i.e., sex and hemisphere) (*Figure 5*; *Figure 5—source data 1*).

**Table 1.** Association between polygenic scores (PGSs) and hippocampal volume (HV).
Linear models were built for HV (left; right; bilateral) using PGS across cohorts (male; female; both) at three representative p-value thresholds (1E-7; 0.01; 1). p-Values of the slope were significant across all categories, with the lowest being associated with the threshold value of 1 in all but a single case (both/right). Variance explained ($R^2$) increased from left to right to bilateral volumes and increased from female to male to both.

| Gender | PGS threshold | LEFT | | | RIGHT | | | BILATERAL | | |
|---|---|---|---|---|---|---|---|---|---|---|
| | | Slope (×10$^{-2}$) | p-Value | $R^2$ | Slope (×10$^{-2}$) | p-Value | $R^2$ | Slope (×10$^{-2}$) | p-Value | $R^2$ |
| | 1E-7 | 10 | 1.8E-46 | 13% | 9.4 | 2.4E-45 | 14% | 11 | 1.4E-51 | 15% |
| | 0.01 | 8.2 | 2.7E-26 | 13% | 7.6 | 1.0E-27 | 13% | 8.7 | 3.2E-30 | 14% |
| FEMALE | 1 | 11 | **9.4E-54** | 13% | 9.62 | **1.5E-48** | 14% | 11 | **1.6E-57** | 15% |
| | 1E-7 | 8.2 | 1.4E-35 | 18% | 7.5 | 2.6E-35 | 18% | 9.2 | 4.1E-40 | 20% |
| | 0.01 | 7.8 | 3.8E-29 | 18% | 6.8 | 3.8E-27 | 18% | 8.6 | 7.8E-32 | 20% |
| MALE | 1 | 9.4 | **3.2E-48** | 18% | 8.0 | **4.7E-43** | 18% | 10 | **9.1E-52** | 20% |
| | 1E-7 | 8.4 | 8.1E-90 | 25% | 7.9 | **6.4E-93** | 26% | 9.3 | 3.1E-103 | 28% |
| | 0.01 | 7.4 | 9.3E-54 | 24% | 6.7 | 3.3E-53 | 26% | 8 | 2.3E-60 | 28% |
| BOTH | 1 | 9.6 | **2.1E-99** | 25% | 8.3 | 1.8E-89 | 26% | 10 | **7.5E-107** | 28% |

Slope = beta coefficient for PGS in the linear mode; p-value for the slope; $R^2$=variance explained by the linear model.

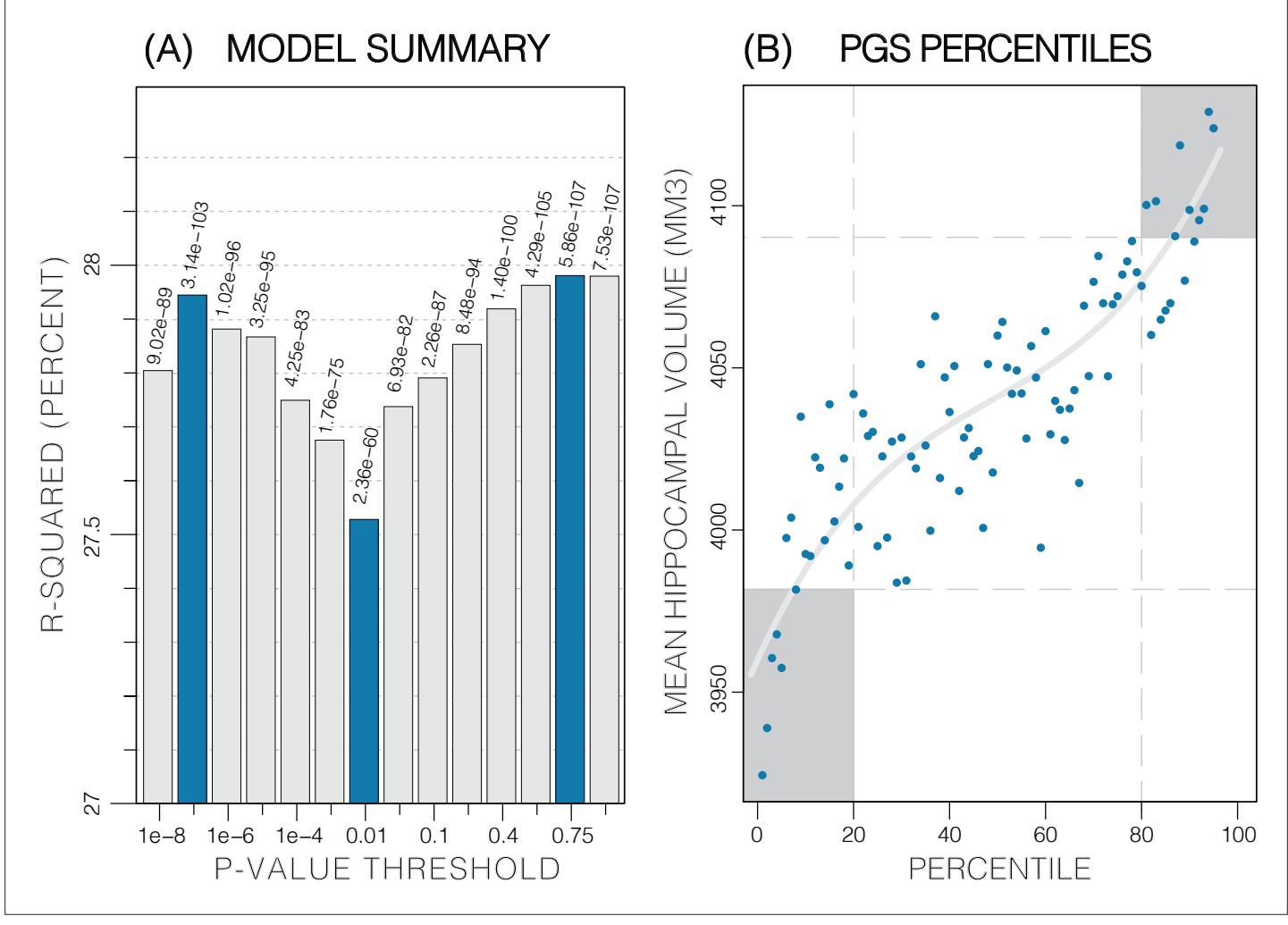

**Figure 3.** Summary of polygenic score (PGS) models. Polygenic risk score in models of mean hippocampal volume (HV) across both sexes. (**A**) $R^2$ of linear models across increasing p-value thresholds. All models are of bilateral HV and account for age, sex, and top 10 genetic principal components. The minimum $R^2$ on the y-scale is the $R^2$ of the models without any PGS. (**B**) Distribution of mean HV across percentiles of PGS. Excluding the top and bottom 20% of percentiles reduces the variance by 49% (darker grey areas). Fitting a cubic polynomial to the means produces the grey line.

The online version of this article includes the following source data and figure supplement(s) for figure 3:

**Source data 1.** Summary of PGS vs HV regression models.

**Figure supplement 1.** Summary of polygenic scores (PGSs) based on hippocampal volume (HV) genome-wide association study (GWAS) in UK Biobank (UKB) samples.

**Figure supplement 2.** Summary of polygenic scores (PGSs) and models based on hippocampal volume (HV) genome-wide association study (GWAS) and Alzheimer's Disease Neuroimaging Initiative (ADNI) samples.

## Longitudinal evaluation

We also investigated whether genetically adjusted nomograms provided additional accuracy in distinguishing stable (n=299) from MCI-to-AD progressing subjects (n=83). With the non-adjusted nomogram, progressing MCI participants had a mean HV percentile of 11% and stable participants had 29% (*Figure 6*). Using the genetically adjusted nomograms, they had 10% and 28%, respectively. Cox proportional hazards models of percentiles obtained using both nomograms revealed little difference between the two in terms of clinical conversion: both models resulted in a hazard ratio of 0.97 for percentile in nomogram (beta of –0.03 at p-value<4.87E-08); confirming that participants within lower HV percentiles where more likely to convert earlier.

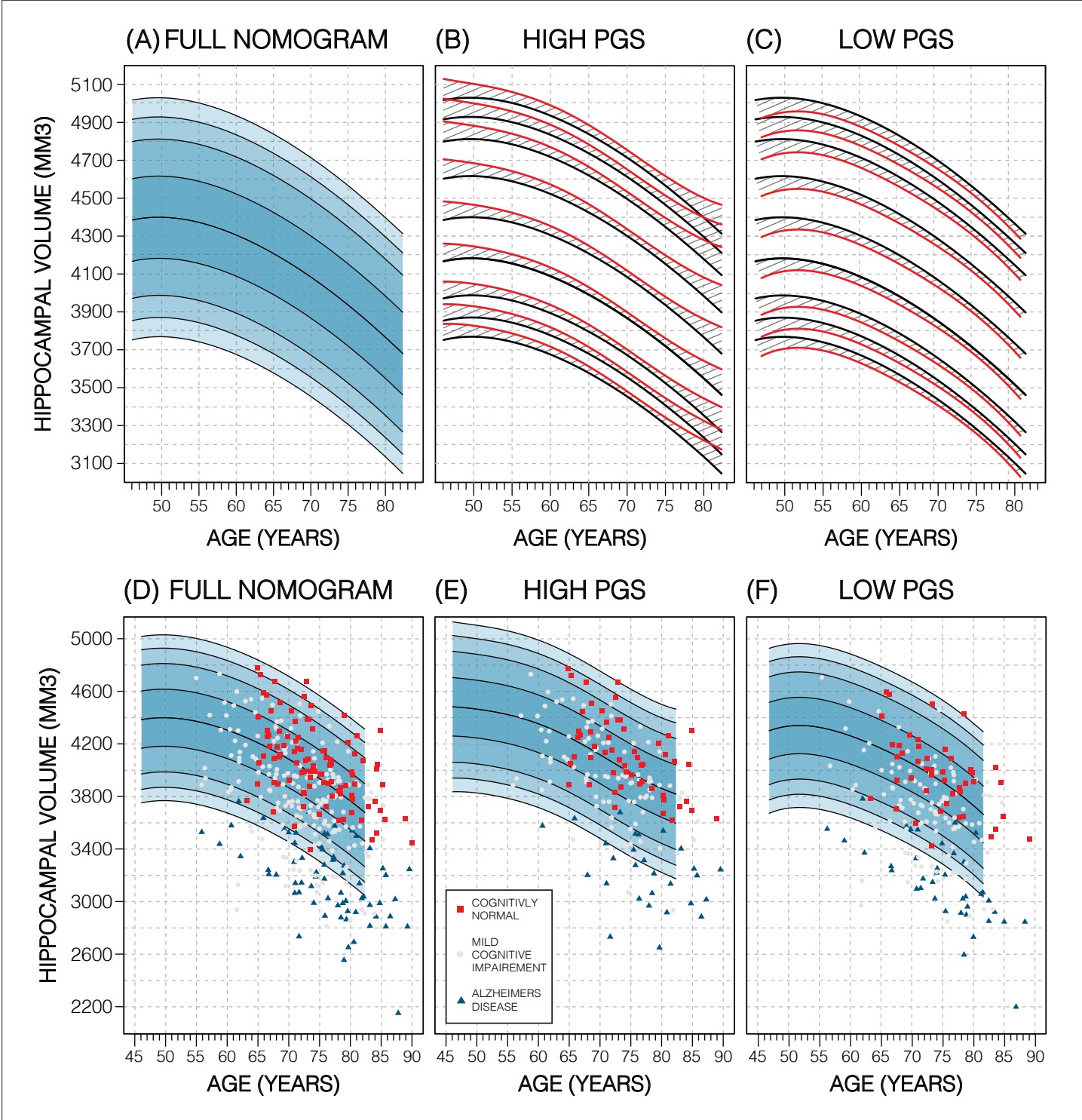

**Figure 4.** Genetically adjusted nomograms. Results of genetic adjustment in bilateral male hippocampal volume (HV). (**A, D**) Nomograms of bilateral HV generated from all male UK Biobank (UKB) samples overlayed with male Alzheimer's Disease Neuroimaging Initiative (ADNI) samples. Cognitively normal (CN) samples (red squares) centre around the 50th percentile, Alzheimer's disease (AD) samples (turquoise triangles) lie mostly below the 2.5th percentile, and mild cognitive impairment (MCI) samples (grey circles) span both regions. (**B, E**) Nomograms generated using only high polygenic score (PGS) samples (top 30%) was shifted upwards (red lines) compared to the original (black lines) by an average of 50 mm³ (1.2% of mean HV). Plotting the high PGS ADNI samples (top 50%) slightly improves intra-group variance. (**C, F**) Similar results are seen in low PGS samples. Note, the black lines in panels (**B, C**) are the same as the nomogram in panel (**A**) and similarly the red lines in panel (**B, C**) are same as the nomogram in panels (**E, F**).

The online version of this article includes the following figure supplement(s) for figure 4:

*Figure 4 continued on next page*

**Figure supplement 1.** Genetically adjusted nomograms.

**Figure supplement 2.** Nomograms generated with the sliding window method (SWM) by stratifying the sample set based on polygenic scores (PGSs).

**Figure supplement 3.** Training data ridge plots.

## Discussion

We hypothesized that inclusion of genetic information associated with regional brain volume may substantially affect normative models. Indeed, the PGS for HV was significantly positively correlated with estimated HV from magnetic resonance imaging (MRI); translating into a shift of around 100 mm$^3$ in nomograms based on PGS stratification (high vs. low PGS). Importantly, this magnitude corresponds to ~3 years' worth of HV loss during normal aging for a 65-year-old. While previous studies have examined the impact of disease-associated variants, such as *APOE* status, on HV (*Ching et al., 2020*; *Veldsman et al., 2021*), our study relied on genetic variants influencing HV in healthy subjects. This is an important difference: the *APOE* genotype is associated with present or future AD status rather than having a direct influence on HV in healthy populations. Indeed, GWASs of the hippocampus that exclude dementia patients find little influence of AD-associated SNPs (*Hibar et al., 2017*). By design, nomograms are intended to model healthy progression and to simplify spotting disease-related outliers. Therefore, in theory, accounting for the genetics of healthy variation in HV should enable earlier detection of AD-related HV decline aging individuals. Conversely, stratifying by *APOE*-e4 status when creating HV nomograms charts the different HV trajectories among *APOE* genotypes, however, at the same time masks the pathological decline and thus will theoretically decrease the sensitivity to HV decline.

Subjects with extreme PGS account for significant amounts of the variance as indicated by the S-shape in the quantile plots (e.g., *Figure 3*). This has been observed in other PGS-trait combinations (*Axelrud et al., 2018*; *Escott-Price et al., 2015*; *Ranlund et al., 2018*). Furthermore, we found similar $R^2$ values across all PGSs (±0.05 $R^2$) with two peaks at thresholds of 1E-7 and 0.75. This reflects two types of genetic effects: the first is that few SNPs account for a substantial portion of the total variance in HV because of their high effect size (oligogenic effect) and the second is the combined effect of all common genetic variants on HV (polygenic effect). This type of effect has been reported in other studies of dementia (*Bis et al., 2012*).

In addition to demonstrating the clear effect of genetics on normative models, we have shown GPR to be effective for estimating nomograms. Using a model-based method allows us to generate accurate nomograms across the entire age range of the dataset. In fact, our GPR model can potentially be extended a few years beyond those limits (*Figure 2—figure supplement 1*). In comparison, the SWA nomograms age range is reduced by 20 years compared to the range of the training because of the required smoothing. Thus, compared to the SWA, we extended the age range forwards by 10 years, bringing it out to 82 years of age, which is relevant for AD where most patients display symptoms at around age 65–75 (*Rabinovici, 2019*; *Mendez, 2017*). While some methods like LOESS regression can be used to mitigate this need (*Bethlehem et al., 2020*), the GPR's model-based approach does not need smoothing to begin with. However, there is a possibility that our results suffer from edge effects. For example, we suspect that the peak noted in the male nomogram is likely due to under-sampling in the younger participants. We found that building nomograms is data efficient: with the SWA, using around 17% (3000 samples) of training samples generated nomograms that were on average only 0.4% of average HV (19 mm$^3$) different to those generated by the full training

**Table 2.** Results of ANOVA tests of UK Biobank (UKB) hippocampal volume (HV) percentiles produced with genetically adjusted and unadjusted nomograms.

| SEX | STRATA | DF | SUM SQ | F-VALUE | p-VALUE |
|---|---|---|---|---|---|
| | HIGH | 1 | 18,786 | 22.84 | 1.8E-06 |
| MEN | LOW | 1 | 16,407 | 19.96 | 8.04E-06 |
| | HIGH | 1 | 27,068 | 32.92 | 9.97E-09 |
| WOMEN | LOW | 1 | 30,103 | 36.94 | 1.28E-09 |

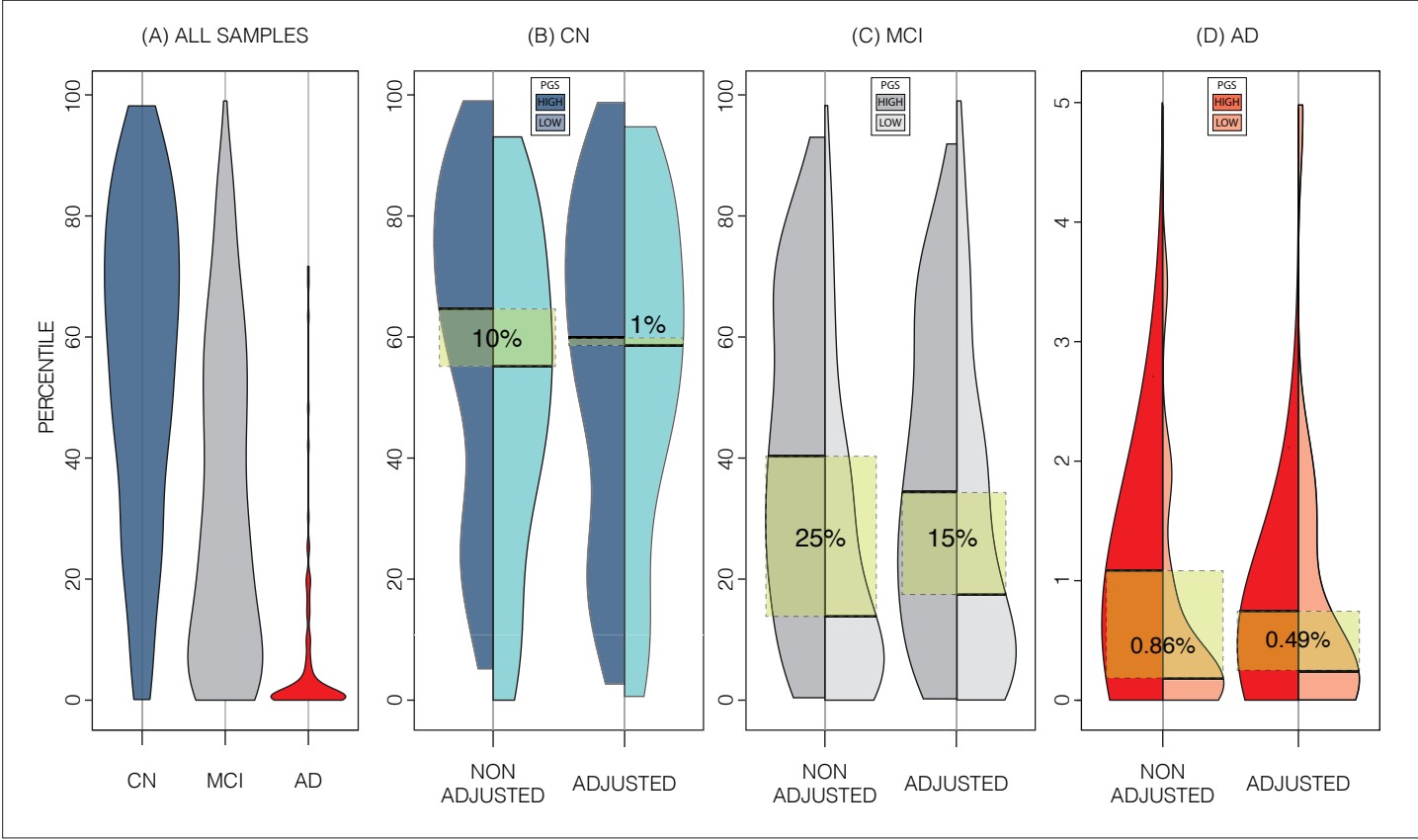

**Figure 5.** Alzheimer's Disease Neuroimaging Initiative (ADNI) dataset percentiles in genetically adjusted/non-adjusted nomograms. Plotting the percentile distribution of the different diagnostic groups across adjusted and non-adjusted nomograms reveals that genetic adjustment increases group cohesiveness. (**A**) The percentile distributions of the different diagnostic groups against the non-adjusted nomograms. (**B**) In cognitively normal (CN) samples for example, when plotting against the non-adjusted nomogram (left adjoined boxplots), the median percentile of the top 30% of samples (darker turquoise) was 65%, while the median for the lower 30% of samples (lighter turquoise) was 54%. When using the genetically adjusted nomogram instead (right adjoined boxplots), those median percentiles become 60% and 59% respectively, a 90% relative reduction. Similar results can be seen with mild cognitive impairment (MCI) (**C**) and Alzheimer's disease (AD) (**D**) samples, with 60% and 56% relative reduction, respectively.

The online version of this article includes the following source data for figure 5:

**Source data 1.** Summary of average percentiles across ADNI strata and UKB nomograms.

set. GPR nomograms achieved the same level of accuracy with only 5% (900 samples) of the dataset (*Figure 2—figure supplement 3*).

Using PGS improves the normative modelling in an independent dataset. In ADNI genetic adjustment reduced the percentile gap between similarly diagnosed subjects with genetically predicted high and low HV. The impact of the PGS adjusted model on CN samples was greater than on MCI or AD samples. Genetic adjustment centred the CN samples closer to the 50th percentile. As the effect of building separate nomograms was to mitigate the impact of genetic variability on HV it was not surprising that this effect did not carry over to MCI and AD subjects, likely because the pathological effect of AD on HV (~804 mm³ or 6.4% volume loss) far exceeds the shift in nomograms achieved with genetic adjustment (~100 mm³ or 0.8% of mean HV). Other studies have found that annual HV loss in CN subjects was between 0.38% and 1.73% (*Scahill et al., 2003*; *Leong et al., 2017*; *Jack et al., 2000*; *Mori et al., 2002*; *Risacher et al., 2010*). Using the nomograms from our work, genetic adjustment corresponds to ~3 years of normal aging for a 65-year-old. However, despite this sizable effect, genetically adjusted nomograms did not provide additional insight into distinguishing MCI subjects that remained stable or converted to AD. Nonetheless, the added precision may prove more useful in early detection of deviation among CN subjects, for instance in detecting subtle HV loss in individuals with presymptomatic neurodegeneration.

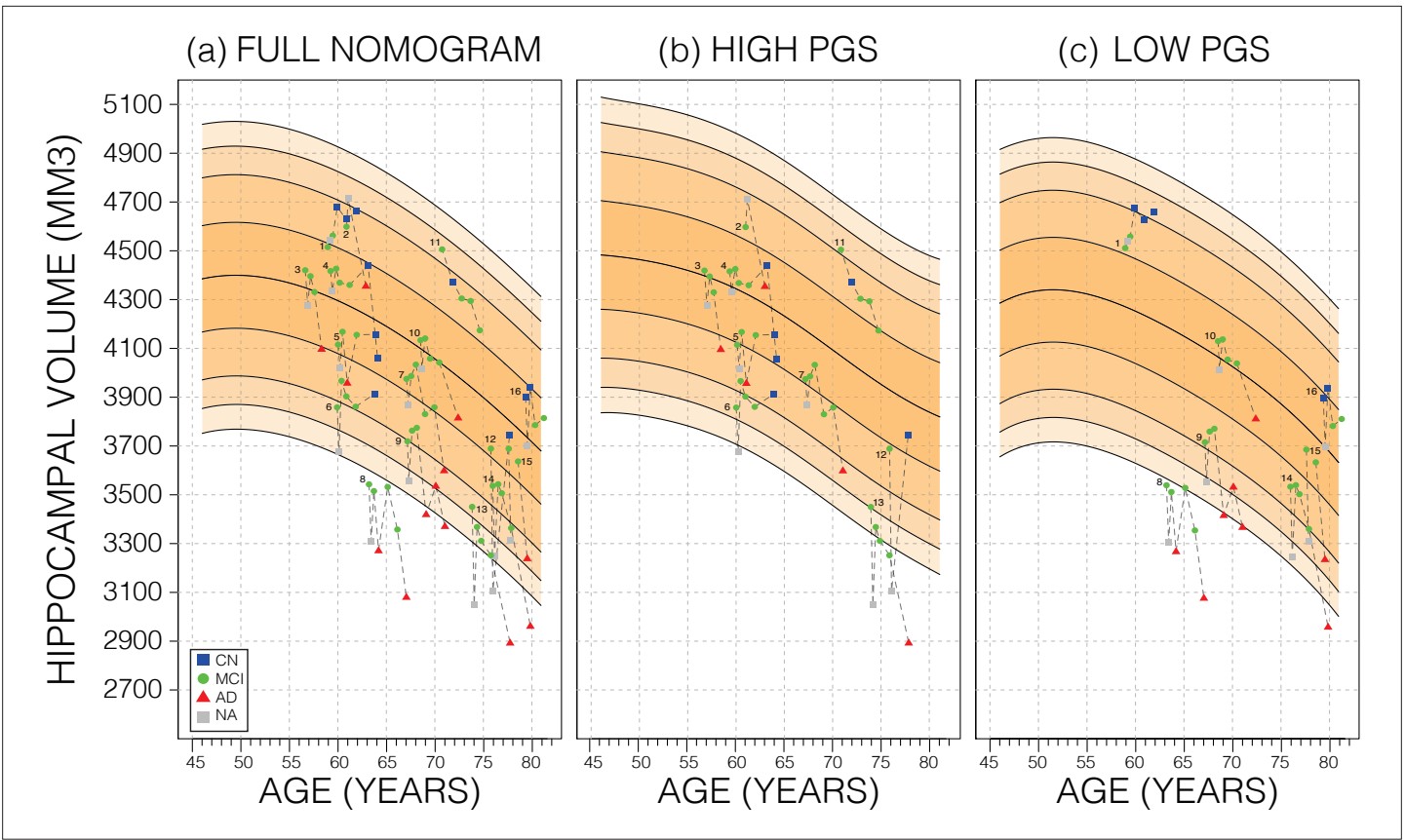

**Figure 6.** Longitudinal analysis. A selection of mild cognitive impairment (MCI) samples longitudinal data plotted against nomograms of male mean hippocampal volume (HV). (**a**) All selected samples plotted against a non-adjusted nomogram. Lines connect visits of the same sample with diagnosis at each visit shown: cognitively normal (CN) as blue squares; MCI as green dots, Alzheimer's disease (AD) as red triangles, and no diagnosis (NA) as grey squares. (**b**) Samples from (**a**) with high polygenic scores (PGS) plotted against a nomogram generated from high PGS CN samples in UK Biobank (UKB). (**c**) Equivalent result for low PGS samples from (**a**). For all sub-figures, the black lines – from top to bottom – represent the 2.5%, 5%, 10%, 25%, 50%, 75%, 90%, 95%, and 97.5% quantiles, respectively.

While this study has shown the significant impact of PGSs on HV nomograms, we have identified areas for improvement. Integrating the PGSs into the GP models would remove the need for stratification and allow for more adjustment precision, however, PGSs are prone to 'site' effects depending on the method and quality of genotyping and imputation. Consequently, using the 'raw' PGSs in predictive models presents its own challenges. Also, the PGSs used in this study were based on a GWAS of average bilateral HV in both male and female participants. Previous studies have shown a significant difference between these groups (*Nobis et al., 2019*), and nomograms estimated for these separate groups are distinct (*Schmidt et al., 2018*; *Khlif et al., 2019*; *Pardoe et al., 2009*; *Figure 2*). Therefore, using separate GWASs for each of these strata would potentially give the PGSs more accuracy. A second limitation of this study is the reliability of HV estimates. There is a significant difference between manual and automated segmentation of the hippocampus (*Schmidt et al., 2018*; *Khlif et al., 2019*; *Pardoe et al., 2009*) more so than other brain regions (*Keller et al., 2012*; *Buser et al., 2020*), and FreeSurfer is known to consistently overestimate HV (*Perlaki et al., 2017*). Therefore, other brain regions with higher SNP heritability like the cerebellum or whole brain volume (*Zhao et al., 2019*) may show more sensitivity on nomograms. Moreover, a recent study of PGS uncertainty revealed large variance in PGS estimates (*Ding et al., 2020*), which may undermine PGS-based stratification; hence a more sophisticated method of building PGS or stratification may improve results further. Finally, while NeuroCombat has been shown to remove most site effects, some may remain and still need to be adjusted for (*Stamoulou et al., 2021*).

In conclusion, our study demonstrated that PGS for HV was significantly positively correlated with HV, translating into a shift in the nomograms corresponding to ~3 years' worth of normal aging HV

loss for a 65-year-old. We have additionally shown that this effect can be observed in an independent dataset. And while more work in this direction is needed, successful integration of polygenic effects on multiple brain regions may help improve the sensitivity to detect early disease processes.

# Materials and methods

## Datasets

Data from a total of 39,664 subjects (18,718 female) aged 44–82 were obtained from the UKB (application number 65299) with available genotyping and imaging data. Imaging and genetic protocols are described in *Bycroft et al., 2018*, and *Miller et al., 2016*, respectively. Briefly, for this analysis we used HV estimated with FreeSurfer (*Fischl, 2012*) at the initial imaging visit. The dataset preparation followed the process described by *Nobis et al., 2019*. To ensure nomograms represent the spectrum of healthy aging, subjects were excluded based on history of neurological or psychiatric disorders, head trauma, substance abuse, or cardiovascular disorders. Furthermore, to control for population-level genetic heterogeneity, only subjects with 'British' ethnic backgrounds were considered. The dataset was then stratified by self-reported sex. HV outliers were excluded using mean absolute deviation with a threshold of 5.0. Subjects' intracranial volume (ICV) was derived by using the volumetric scaling from T1 head image to standard space. Finally, ICV and scan date were linearly regressed out of the HVs.

For an application dataset, we used the ADNI database (http://adni.loni.usc.edu/) (*Petersen et al., 2010*). The ADNI was launched in 2003 as a public-private partnership, led by Principal Investigator Michael W Weiner, MD. The primary goal of ADNI has been to test whether serial MRI, positron emission tomography, other biological markers, and clinical and neuropsychological assessment can be combined to measure the progression of MCI and early AD. A total of 1001 ADNI subjects (445 male) aged 55–95 were included in this analysis. Imaging and genetic protocols are described by *Saykin et al., 2010*, and by *Jack et al., 2008*, respectively. Briefly, we obtained HVs estimated with FreeSurfer v5.1. Subjects were excluded based on HV quality scores and based on genetic ancestry (i.e., restricted to self-reported white non-Hispanic ancestry). As with UKB, estimated volumes were stratified by sex, and ICV and scan date were regressed out of HV estimates. Finally, we used NeuroCombat (*Fortin et al., 2018*) to adjust across ADNI sites and harmonize the volumes with the UKB dataset. To do this we modelled 58 batches (UKB data as one batch and 57 ADNI sites as separate batches) and added ICV, sex, and diagnosis (assigning all UKB as healthy and using the diagnosis columns in ADNI) to retain biological variation. Demographics were obtained from the ADNIMERGE table (date accessed: 19 June 2020). Furthermore, we used genotyping data of ADNI subjects pre-processed as previously described by *Scelsi et al., 2018*.

## Sliding window approach

As a baseline, we generated nomograms using the SWA described by *Nobis et al., 2019*. Briefly, we sorted UKB samples by age, and formed 100 quantile bins, each containing 10% of the samples. This means that neighbouring bins had a 90% overlap. For example, if we had 5000 samples, each bin contained 500 samples and consecutive bins were shifted by 50 samples. Thus, bin number 4 would start at index 151. Then, within each bin, the 2.5%, 5%, 10%, 25%, 50%, 75%, 90%, 95%, and 97.5% quantiles were calculated. The quantiles were then smoothed with a Gaussian kernel of width 20. The smoothing was needed because towards the ends of the data, the sliding windows approach becomes sensitive to noise.

## Gaussian process regression

Our proposed approach uses GPR to build nomograms. Briefly, a GP is a probability distribution over possible functions that fit a set of points (*Rasmussen and Williams, 2006*; *Wang, 2021*). In our application it is a distribution of possible 'HV trajectories across age'. The GPR models were trained with the laGP (*Gramacy, 2016*) R library, which implements a local approximation method that allows large datasets to be trained on consumer grade machines. We applied the commonly used squared exponential covariance kernel function:

$$K\left(x_1, x_2\right) = \sigma^2 e^{\frac{-(x_1 - x_2)^2}{2L^2}},$$

where $x_1$ and $x_2$ are any two age values from the training set. The kernel function is hyper-parameterized by a vertical scale ($\sigma$) and a length scale ($L$), which, following initialization, are fitted using maximum likelihood estimation. The vertical scale is initialized to the mean HV of all samples, and the length scale is initialized to mean age difference between all samples. We trained models of left, right, and mean HV for each sex. Thanks to their probabilistic formulation, GP models naturally provide a standard deviation from which quantiles can be easily computed. After training, we generated models for ages 45–82 by increments of 0.25 years, and quantile curves at 2.5%, 5%, 10%, 25%, 50%, 75%, 90%, 95%, and 97.5%. The UKB dataset was used to train the models and the ADNI dataset was used to test them. For all GPR models, we only tested the ADNI samples that lay within the age range of each model respectively.

## PGS for HV

A PGS is a sum of the impact of a selection of genetic variants on a trait, weighted by the allele count. That is:

$$PGS = \sum_{\forall i \in SNPs} ES_i * C_i,$$

where ($ES_i$) is the effect size (e.g., beta or log(odds) ratio from GWAS summary statistics), and ($C_i$) is the allele count of SNP $i$ in the subject (either 0, 1, or 2). Thus, computing PGSs requires SNP-level genetic data. Using a previously reported GWAS of mean bilateral HV using 26,814 (European) subjects from the ENIGMA study (*Hibar et al., 2017*), we built a PGS for HV with PRSice v2 (*Choi and O'Reilly, 2019*). For both UKB and ADNI, we filter for minor allele frequency of 0.05, genotype missingness of 0.1, and clumping at 250 kb; after which we were left with 70,251 potential SNPS to include for UKB and 114,812 for ADNI. The most widely applied strategy for SNP selection is p-value thresholding. We generated PGSs at 14 p-value thresholds (1E-8, 1E-7, 1E-6, 1E-5, 1E-4, 1E-3, 0.01, 0.05, 0.1, 0.2, 0.4, 0.5, 0.75, 1). These thresholds produced a range of PGSs comprising as little as six SNPs (p-value cut-off at 1E-8) to all available SNPs (p-value cut-off at 1.0). Model fit is then checked by regressing HV against these PGSs while accounting for age, age$^2$, sex, ICV, and 10 genetic principal components.

## Genetically adjusted nomograms

Given the high heritability of HV we investigated whether nomograms can be genetically adjusted. Specifically, we used the top and bottom 30% samples by PGS (at p-value<0.75 threshold) separately to build genetically adjusted nomograms. We found that using a 30% cut-off provided a balance of training size and performance (*Figure 2—figure supplement 4*). Thus, PGS provided us with a way to place new samples in their 'appropriate' nomogram. For instance, within the ADNI dataset we generated PGSs and split the top and bottom (i.e., high and low expected HV, respectively) to test against genetically adjusted UKB nomograms. To evaluate the impact of genetic adjustment, we perform a series of ANOVA tests across adjusted nomograms. For example, we performed an ANOVA test of the HV percentiles of the top 30% UKB samples in the unadjusted than the adjusted nomograms. We did the same for bottom 30% and for men and women. To assess the specificity of the HV-based PGS, we performed this genetic adjustment using PGSs of ICV and AD based on previously reported GWASs (*Adams et al., 2016*; *Lambert et al., 2013*).

## Longitudinal analysis

As nomograms are often used to track progression, we examined the impact of the genetically adjusted nomograms on prospective longitudinal data. To this end, we analysed patients from the ADNI cohort that were initially diagnosed as MCI and either converted to AD (progressor) or remained MCI (stable) within 5 years of follow-up. We tested whether the PGS-adjusted nomograms improved the separation between stable and progressor patients using Cox proportional hazards models while accounting for sex and age.

## Code and data availability

The scripts and code used in this study have been made publicly available and can be found at: https://github.com/Mo-Janahi/NOMOGRAMS; *Janahi, 2021*. All underlying data, and derived quantities, are available by application from the UKB at http://www.ukbiobank.ac.uk, and by application

from ADNI at http://adni.loni.usc.edu/data-samples/access-data/. Summary statistics from all GWAS described in this paper are available from the NHGRI-EBI GWAS catalog, study numbers: GCST003834, GCST002245, and GCST003961. URL: https://www.ebi.ac.uk/gwas/studies/.

## Acknowledgements

AA holds an MRC eMedLab Medical Bioinformatics Career Development Fellowship. This work was supported by the Medical Research Council (grant number MR/L016311/1). This work was supported in part by Sidra Medicine, Qatar. LMA was supported by the National Institute of Biomedical Imaging and Bioengineering of the National Institutes of Health under Award Number P41EB015922 and by the National Institute on Aging of the National Institutes of Health under Award Number P30AG066530. JMS acknowledges the support of the UCL/H NIHR Biomedical Research Centre. This work is supported by the EPSRC-funded UCL Centre for Doctoral Training in Intelligent, Integrated Imaging in Healthcare (i4health) (EP/S021930/1).

Data used in preparation of this article were obtained from the ADNI database (http://adni.loni.usc.edu/). Data collection and sharing for this project was funded by the ADNI (National Institutes of Health Grant U01 AG024904) and DOD ADNI (Department of Defense award number W81XWH-12-2-0012). ADNI is funded by the National Institute on Aging, the National Institute of Biomedical Imaging and Bioengineering, and through generous contributions from the following: AbbVie, Alzheimer's Association; Alzheimer's Drug Discovery Foundation; Araclon Biotech; BioClinica, Inc; Biogen; Bristol-Myers Squibb Company; CereSpir, Inc; Cogstate; Eisai Inc; Elan Pharmaceuticals, Inc; Eli Lilly and Company; EuroImmun; F Hoffmann-La Roche Ltd and its affiliated company Genentech, Inc; Fujirebio; GE Healthcare; IXICO Ltd.; Janssen Alzheimer Immunotherapy Research & Development, LLC.; Johnson & Johnson Pharmaceutical Research & Development LLC.; Lumosity; Lundbeck; Merck & Co., Inc; Meso Scale Diagnostics, LLC.; NeuroRx Research; Neurotrack Technologies; Novartis Pharmaceuticals Corporation; Pfizer Inc; Piramal Imaging; Servier; Takeda Pharmaceutical Company; and Transition Therapeutics. The Canadian Institutes of Health Research is providing funds to support ADNI clinical sites in Canada. Private sector contributions are facilitated by the Foundation for the National Institutes of Health (https://www.fnih.org/). The grantee organization is the Northern California Institute for Research and Education, and the study is coordinated by the Alzheimer's Therapeutic Research Institute at the University of Southern California. ADNI data are disseminated by the Laboratory for Neuro Imaging at the University of Southern California. As such, the investigators within the ADNI contributed to the design and implementation of ADNI and/or provided data but did not participate in analysis or writing of this report. Data used in the preparation of this article were obtained from the ADNI database (http://adni.loni.usc.edu/). As such, the investigators within the ADNI contributed to the design and implementation of ADNI and/or provided data but did not participate in analysis or writing of this report. A complete listing of ADNI investigators can be found at: http://adni.loni.usc.edu/wp-content/uploads/how_to_apply/ADNI_Acknowledgement_List.pdf.

## Additional information

### Funding

| Funder | Grant reference number | Author |
| --- | --- | --- |
| Medical Research Council | MR/L016311/1 | Andre Altmann |
| National Institute of Biomedical Imaging and Bioengineering | P41EB015922 | Leon Aksman |
| National Institute on Aging | P30AG066530 | Leon Aksman |

The funders had no role in study design, data collection and interpretation, or the decision to submit the work for publication.

## Author contributions
Mohammed Janahi, Resources, Data curation, Software, Formal analysis, Investigation, Visualization, Methodology, Writing - original draft, Writing - review and editing; Leon Aksman, Conceptualization, Writing - review and editing; Jonathan M Schott, Supervision, Writing - review and editing; Younes Mokrab, Writing - review and editing; Andre Altmann, Conceptualization, Resources, Supervision, Project administration, Writing - review and editing

## Author ORCIDs
Mohammed Janahi http://orcid.org/0000-0001-7442-2298
Andre Altmann http://orcid.org/0000-0002-9265-2393

## Decision letter and Author response
Decision letter https://doi.org/10.7554/eLife.78232.sa1
Author response https://doi.org/10.7554/eLife.78232.sa2

## Additional files

### Supplementary files
• MDAR checklist

### Data availability
The scripts and code used in this study have been made publicly available and can be found at: https://github.com/Mo-Janahi/NOMOGRAMS, (copy archived at swh:1:rev:2522548b320b3a9859a539bd7b06768dffb38f7e).

The following previously published datasets were used:

| Author(s) | Year | Dataset title | Dataset URL | Database and Identifier |
|---|---|---|---|---|
| Adams HH | 2016 | Novel genetic loci underlying human intracranial volume identified through genome-wide association | https://www.ebi.ac.uk/gwas/studies/GCST003834 | EMBL-EBI, GCST003834 |
| Lambert JC | 2013 | Meta-analysis of 74,046 individuals identifies 11 new susceptibility loci for Alzheimer's disease | https://www.ebi.ac.uk/gwas/studies/GCST002245 | EMBL-EBI, GCST002245 |
| Hibar DP | 2017 | Novel genetic loci associated with hippocampal volume | https://www.ebi.ac.uk/gwas/studies/GCST003961 | EMBL-EBI, GCST003961 |

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
