## [Editor Report]

This manuscript considers whether genetic information can improve the clinical utility of population norms derived from brain imaging data. The authors propose to incorporate polygenic scores into normative models of hippocampal volume to improve predictions of neurodegenerative disease. This approach is elegantly demonstrated in this manuscript and may be useful for clinical translation of population neuroimaging.

---

## [Decision Letter]

**Decision letter after peer review:**

Thank you for submitting your article "Nomograms of Human Hippocampal Volume Shifted by Polygenic Scores" for consideration by *eLife*. Your article has been reviewed by 3 peer reviewers, and the evaluation has been overseen by a Reviewing Editor and Jeannie Chin as the Senior Editor. The following individuals involved in the review of your submission have agreed to reveal their identity: Andre F Marquand (Reviewer #1); Richard AI Bethlehem (Reviewer #2).

Essential revisions:

1. The authors argue that the use of Gaussian processes enables predictions outside the age range that the model is trained on. This would enable a model trained on UK Biobank to be applied to the ADNI dataset. The reviewers express scepticism about this claim and request further evidence for its validity.

2. The reviewers note the need for a more rigorous quantification and/or detailed presentation of the amount of improvement provided by the genetically informed models, and of the quality of fit.

3. Please provide a more in-depth consideration of potential sources of confounds, particularly site effects for the ADNI data.

4. It would appear that the UK Biobank and ADNI datasets deviate in several key properties relevant to the modelling. Some investigation into the implications of this would considerably strengthen the paper.

5. All reviewers request more in-depth consideration of the details of modelling including:

5a. How to deal with non-Gaussianities in the data;

5b. How the models are trained in practice (e.g., test/train split, initialisation);

5c. Effect of selection of subjects with high vs low polygenic score, and the application.

*Reviewer #2 (Recommendations for the authors):*

This is a very interesting and well-written paper and I only have some small suggestions and comments related to mainly the methods and results.

As noted in the public review I think the section on GPR methodology could do with a lot more detail. Such as but not limited to discussion of:

– What does the test train split look like;

– Does the initiation of GPR at the mean pose an issue for data that may not be normally distributed at a given age (I assume not since these are Gaussian processes)?

– Would there be an issue if the variability of a given phenotype (HV in this case) varies across the lifespan? In our own recent work, we observed that certainly for many cortical phenotypes there is an enormous change in variability across the lifespan and so would not expect to see nice parallel quantiles/centile lines such as the ones produced by GPR.

I wasn't sure why the sliding window approach could not be closer to the actual range of the data with perhaps some kind of padding approach that for example, LOESS allows you to use. So I think it is a bit of an oversell of GPR to say it extends the age range as it doesn't extend it really beyond the data that is actually available. I don't think this paper needs to emphasize that as an improvement or to make that contrast so explicit.

The results themselves could perhaps be further strengthened with a visualisation of centile/quantile distributions in the ADNI dataset as they are discussed quite a lot in the results and since these are all effectively age-normalised scores can easily be put into a box/violin/raincloud plot. I think that would also satisfy my curiosity about the skewness of some of the results as it is noted that in the original model AD patient has a mean quantile of 4% with an SD of 10%, so this must be a highly skewed distribution? If so, then maybe it's more appropriate to report the medians of each group.

Finally, it was interesting to see that the CN group in the ADNI dataset had a mean quantile around 41% which to me would suggest that this dataset as a whole is somewhat offset from the UK BioBank sample as a perfectly "normal" other group should hover around the 50% by definition. While the PGS weighting seems to normalise this somewhat it did make we wonder whether there should be some kind of a prior normalisation or general study weighting to apply a UKB-derived model to a new dataset? On a related note: how did the authors deal with the enormous site-level variation within ADNI?

*Reviewer #3 (Recommendations for the authors):*

1) As already mentioned in my public review, my main concern is the applicability of the model to the ADNI dataset. The model can clearly not be extended outside of the age range when considering younger ages. I must admit that for the ADNI cohort / older ages the model seems more reliable based on what we know from the literature but that is not sufficient. I am not sure how to solve this problem, other than adding the CN subjects from ADNI to the creation of the nomograms, although that could lead to a whole range of other harmonisation problems. Another option would be to limit the analysis to include only those subjects that are within the age range.

2) Is it possible to quantify the improvement when adding the genetic information to the nomograms? See also point 6) below.

3) Line 152: "… and scan date were regressed out of the TVs" How? Is it reasonable to assume that the scanner drift is linear (the Github scripts seem to suggest this is what was modelled) but this also suggests e.g. no scanner updates, hardware changes, and so on? Was there also a correction for the different scanners that may have been used (as far as I am aware, UKB has several imaging sites).

4) Line 220: What is the rationale for splitting high-versus low PGS at 30%? What happens at the other thresholds? Why is there a different choice for ADNI?

5) Line 239: The dropout number for HV in ADNI is pretty large and probably non-random. Please comment.

6) What is the meaning of {plus minus}30% in statements like "cognitively normal (CN) participants (n = 225) had a mean bilateral HV percentile of 41% ({plus minus}30%)"? Is it standard deviation/standard error? These errors seem rather large, so that leads me to believe that the e.g. 4% drop could be too small to be meaningful.

7) Discussion, first paragraph: "Therefore, accounting for … " This statement seems to contradict the results. Maybe this discussion is better placed elsewhere.

8) Discussion, second paragraph / Figure 3 / Supplementary FiguresS1/S2 / Supplementary Table S1.

The (supplementary) Figures are very misleading if you compare these with supplementary Table 1: from Table 1 I conclude that every threshold predicts HV about equally well, but the figure suggests otherwise if you do not pay attention to the cut-off in the y-axis. The paragraph in the discussion that describes the so-called bimodal distribution supports this (false) idea and should be removed.

9) Discussion, Line 423: "Therefore, other brain regions with higher heritability like the cerebellum or whole brain volume may show more sensitivity on nomograms." I am somewhat confused about this sentence. Do the authors mean to imply that structures with higher heritability might benefit more from stratifying on PGS? This would only be the case if not only heritability but also SNP heritability should be higher (and the latter also depends on the genetic architecture and discovery sample size).

10) Discussion, final sentence, the brain age gap has not been mentioned in the paper up to this point. While potentially relevant, it is strange to introduce it in the final sentence.

11) Ethics: I would have expected some statements about the use of human data from UKB and ADNI in this paragraph.

12) Supplementary Figure S5: there are people that seem to switch diagnosis from AD back to MCI, this cannot be right?

13) Throughout the paper there are statements like "Importantly, this magnitude corresponds to ~3 years' worth of HV loss during normal aging." This suggests a constant loss over the lifespan (i.e. a linear pattern with age, but the data shows a different pattern. Please rephrase.

14) The (Supplementary) Figures could use a little bit more attention:

- A little bit more information on what is shown in the figures is needed to be able to assess what is displayed; e.g. add abbreviations to the captions, there are no units for some of the axes. None of the nomogram figures have labels for percentile lines, which is essential. Figure S1&S2 please explain the percentile figures.

[Editors' note: further revisions were suggested prior to acceptance, as described below.]

Thank you for resubmitting your work entitled "Nomograms of Human Hippocampal Volume Shifted by Polygenic Scores" for further consideration by *eLife*. Your revised article has been evaluated by Jeannie Chin (Senior Editor) and a Reviewing Editor.

The manuscript has been improved but there are some remaining issues that need to be addressed, as outlined below:

1. Regarding tests for Gaussianity in the UKB samples. We recommend in Figure 4 —figure supplement 3 that the interpretation of the Shapiro-Wilks test is clarified. That is, state explicitly that a given distribution is designated as non-Gaussian if the SW test yields p below some threshold. Also, we believe it is the "Shapiro-Wilk" or "Shapiro-Wilks" test, not "Shapiro-Wilkens".

2. Throughout, the authors use the term "PGS score" which would be written in full as "polygenic score score". We appreciate the awkwardness that sometimes comes with acronyms, but suggest sticking to either "PGSs" or "PG scores".

3. It might be worthwhile adding some discussion regarding Reviewer 1's comments about the potential benefits of directly incorporating PGSs in normative modelling, alongside the challenges that the authors raise in their response letter.

4. It might help readers less familiar with sliding window techniques to be even more explicit about the reason why smoothing restricts the age range. The authors state this but do not note that this is due to "edge effects", in which smoothed sliding window curves become highly sensitive to noisy data at the limits of data ranges.

5. The new results in Figure 5 might be better visualised as violin or raincloud plots. However, we do appreciate that Reviewer 2, who requested this information, did also suggest that boxplots would suffice.

6. Please consider dampening the conclusions ever so slightly. NeuroCombat generally does an excellent job at removing some site related variation, but does not remove the tenacious issue of site effect altogether.

---

## [Author Response]

Essential revisions:1. The authors argue that the use of Gaussian processes enables predictions outside the age range that the model is trained on. This would enable a model trained on UK Biobank to be applied to the ADNI dataset. The reviewers express scepticism about this claim and request further evidence for its validity.

Thanks for raising this point. We agree that the Gaussian processes cannot reliably predict outside the range they have been trained on. Essentially this comment is based on a miscommunication on our side that we have done our best to clear up. To clarify, none of the models in this work are predicting data outside of the range that they are trained on. What we meant to highlight was that the Gaussian Processes have an extended range compared to the Sliding Window method, specifically because the GP models do not require smoothing, and so can use the full age-rage of the training dataset. In the application dataset, ADNI, we only compare the participants who fall within the age-range of the models they are tested against. Hence, we stated that the GP models enable more of the ADNI dataset to be utilized. We provide more details in the response to specific reviewer comments below.

2. The reviewers note the need for a more rigorous quantification and/or detailed presentation of the amount of improvement provided by the genetically informed models, and of the quality of fit.

Metrics and figures for the quality of fit have been added. In brief, the newly produced figure 5 illustrates the improvement in intra-diagnostic group variance achieved by using genetically adjusted nomograms. ANOVA tests have been performed to check the statistical significance of the difference between adjusted and unadjusted nomograms. Shapiro-Wilkens tests of normality have been provided to address the concerns of non-Gaussianities and model fit.

3. Please provide a more in-depth consideration of potential sources of confounds, particularly site effects for the ADNI data.

We agree that multi-site confounds are indeed a concern in neuroimaging. However, in this case, our training was done in one dataset (UKB), where all reasonable attempts were made to minimize site biases. Thus, for the model training we do not anticipate strong site effects. For the application, we do in fact not use a prediction by the GP, but instead we remove the effects of intracranial volume (ICV) and HV. Then we age, HV (and genetics) to convert the HV into a percentile based on the estimated nomograms. However, we agree that site effects remain an issue. Thus, to remove the site effects between the training data (UKB) and the application data (ADNI) we now apply neuroCombat. The new results reflect the better harmonization between datasets.

4. It would appear that the UK Biobank and ADNI datasets deviate in several key properties relevant to the modelling. Some investigation into the implications of this would considerably strengthen the paper.

Indeed, there are various differences between UKB and ADNI starting from MRI acquisition (in some cases field strength) to the software versions used to process the T1w MRIs. Furthermore, ADNI is a cohort enriched for disease observations while UKB is a population cohort. However, our application of NeuroCombat should have removed the majority of technical confounds.

5. All reviewers request more in-depth consideration of the details of modelling including:5a. How to deal with non-Gaussianities in the data;

We apologize for the misunderstanding, the non-Gaussianities seen in the figure reflected the distribution in the ADNI cohort and were driven by disease effects. The UKB data followed a Gaussian distribution. We have added now figures and provided metrics that clear up this confusion and confirm that the data being used is gaussian where it is required to be.

5b. How the models are trained in practice (e.g., test/train split, initialisation);

We have now clarified these details in the methods section. See responses to the reviewers’ comments.

5c. Effect of selection of subjects with high vs low polygenic score, and the application.

We have provided extra figures and extra metrics to address the points raised by the reviewers. Please find the details in the response to reviewers’ comments below.

Reviewer #2 (Recommendations for the authors):This is a very interesting and well-written paper and I only have some small suggestions and comments related to mainly the methods and results.I think the section on GPR methodology could do with a lot more detail. Such as but not limited to discussion of:– What does the test train split look like;

We have added details requested by all reviewers into the GPR methods section (Lines 193-210), for the train/test split we added the following:

Lines 476-478: “The UKB dataset was used to train the models and the ADNI dataset was used to test them.”

– Does the initiation of GPR at the mean pose an issue for data that may not be normally distributed at a given age (I assume not since these are Gaussian processes)?

It does not pose an issue; it is recommended to initialize at the means of the data and then to find bounds for them so that the MSE can test within.

– Would there be an issue if the variability of a given phenotype (HV in this case) varies across the lifespan? In our own recent work, we observed that certainly for many cortical phenotypes there is an enormous change in variability across the lifespan and so would not expect to see nice parallel quantiles/centile lines such as the ones produced by GPR.

This is a very interesting point. Generally, this would not pose an issue for GPR, it gives a semi-independent variance at each age (depending on the vertical hyperparameter), so even if the variance were to change across the life span, it will adjust accordingly. As supporting evidence, note how the GPR matches the SWM (a non-model-based method) in the age-range where they overlap (Figure 2 A-D):

Moreover, compared to the recent lifespan work our focuses on a rather ‘narrow’ window of ~37 years that is dominated by volume decline.

I wasn't sure why the sliding window approach could not be closer to the actual range of the data with perhaps some kind of padding approach that for example, LOESS allows you to use. So I think it is a bit of an oversell of GPR to say it extends the age range as it doesn't extend it really beyond the data that is actually available. I don't think this paper needs to emphasize that as an improvement or to make that contrast so explicit.

We agree that some methods can be used to improve the sliding window method and get it closer the full age-range of the training set, though we maintain that the model-based GPR still has better performance. We have de-emphasized and contextualized the differences between the models where they are mentioned, and we have added to the discussion how the SWM could be improved as well. The specific changes are:

Line 47-48 (in the abstract): “we built HV nomograms using gaussian process regression (GPR), which – compared to a previous method – extended the application age by 20 years, including dementia critical age ranges.”

Lines 119-120 (end of introduction): “We found that genetic adjustment did in fact shift the nomograms and that, because the model requires no smoothing, our GPR nomograms provided an extended age range compared to previous methods.”

Lines 145-147 (SWM vs GPR Results section): “However, GPR nomograms spanned the entire training dataset age range (45-82 years) compared to the SWA (52-72 years). This is primarily because the SWA is a non-model-based approach that requires smoothing, and a gaussian smoothing window of width 20 was used”

Lines 349-354 (Discussion): “In comparison, the SWA nomograms age range is reduced by 20 years compared to the range of the training because of the required smoothing. Thus, compared to the SWA, we extended the age range forwards by 10 years, bringing it out to 82 years old, which is relevant for AD where most patients display symptoms at around age 65-75^4,5^. While some methods like LOESS regression can be used to mitigate this need^6^, the GPR’s model-based approach does not need smoothing to begin with.”

The results themselves could perhaps be further strengthened with a visualisation of centile/quantile distributions in the ADNI dataset as they are discussed quite a lot in the results and since these are all effectively age-normalised scores can easily be put into a box/violin/raincloud plot. I think that would also satisfy my curiosity about the skewness of some of the results as it is noted that in the original model AD patient has a mean quantile of 4% with an SD of 10%, so this must be a highly skewed distribution? If so, then maybe it's more appropriate to report the medians of each group.

Thanks for the suggestion. Indeed, reporting the medians was more appropriate. We have added the requested boxplots as an additional figure (Figure 5) and edited the corresponding paragraph in the Results section (lines 243-257) with these updated results.

We updated the Results section as follows:

Lines 243-257: “In the ADNI dataset we investigated whether the shift in genetically adjusted nomograms could be leveraged for improved diagnosis. Using the non-adjusted nomogram, cognitively normal (CN) participants (n = 225) had a median bilateral HV percentile of 61% (±25% SD), Mild Cognitive Impairment (MCI) participants (n = 391) had 25% (±26% SD), and Alzheimer’s Disease (AD) participants (n = 121) had 1% (±9% SD) (Figure 5). Visual inspection revealed that while CN participants were spread across the quantiles, AD participants lay mostly below the 2.5% quantile, and MCI participants spanned the range of both CN and AD participants (Figure 4). Bisecting the samples by PGS showed that high PGS CN samples had median percentiles of 65% (±27% SD) and low PGS had 54% (±26% SD). When comparing the same samples against the genetically adjusted nomograms instead, high PGS CN samples had 60% (±26% SD) and low PGS had 59% (±26% SD). Thus, reducing the gap between high and low PGS CN participants by 9% (from 10% to 1%; a 90% relative reduction). Similar analysis showed a reduction in MCI participants by 9% (60% relative reduction), and 0.5% (56% relative reduction) in AD participants. The above effects persisted across most strata (i.e., sex and hemisphere) (Figure 5; Figure 5 – Data Source 1).”

Finally, it was interesting to see that the CN group in the ADNI dataset had a mean quantile around 41% which to me would suggest that this dataset as a whole is somewhat offset from the UK BioBank sample as a perfectly "normal" other group should hover around the 50% by definition. While the PGS weighting seems to normalise this somewhat it did make we wonder whether there should be some kind of a prior normalisation or general study weighting to apply a UKB-derived model to a new dataset? On a related note: how did the authors deal with the enormous site-level variation within ADNI?

Thank you for this suggestion. We agree that site effects are a major issue; we have rerun the application experiments after adjusting the ADNI volumes with NeuroCombat. We used one run of NueroCombat to account for both the ADNI-UKB differences and the ADNI site effects. The results did not change significantly, but we have changed all the reported results with the adjusted results. In addition, we noted this in the methods section:

Lines 442-445: Finally, we used NeuroCombat ^1^ to adjust across ADNI sites and harmonize the volumes with the UKB Dataset. To do this we modelled 58 batches (UKB data as one batch and 57 ADNI sites as separate batches) and added ICV, sex, and diagnosis (assigning all UKB as Healthy and using the diagnosis columns in ADNI) to retain biological variation.

Reviewer #3 (Recommendations for the authors):1) My main concern is the applicability of the model to the ADNI dataset. The model can clearly not be extended outside of the age range when considering younger ages. I must admit that for the ADNI cohort / older ages the model seems more reliable based on what we know from the literature but that is not sufficient. I am not sure how to solve this problem, other than adding the CN subjects from ADNI to the creation of the nomograms, although that could lead to a whole range of other harmonisation problems. Another option would be to limit the analysis to include only those subjects that are within the age range.

As mentioned already in response to reviewer #1, this was a miscommunication on our side. We only used the ADNI samples that were within the age range of the models they were being plotted against. The GPR model did not require smoothing at the edges of the age-range and thus can support a wider age range than the SWA. This is why we stated that the extension of the nomograms enabled more of the ADNI dataset to be used, i.e., because otherwise these samples were outside the range of the model and could not be used.

We have changed the following lines in the manuscript to make this idea explicit:

Lines 477-478 (end of GPR methods section): “For both SWM and GPR models, we only tested the ADNI samples that lay within the age range of each model respectively.”

Regarding the accurate extension claim, we have edited the line (411-412) in the discussion so that it now reads:

Lines 347-348 “In fact, our GPR model can potentially be extended a few years beyond those limits”

Thank you for pointing out the discrepancy in the hippocampal growth around 48 with the results by Dima et al. 2022. Although sample sizes between the two studies are similar. The data availability in UKB for ages 45-50 is rather sparse (N<100; see new Figure 4 —figure supplement 3). Thus, the observed growth is likely due to under sampling. The growth effect has been observed in other studies using UKB data^7,8^. We have noted this in the discussion:

Lines 354-356:” However, there is a possibility that our results suffer from edge effects. For example, we suspect that the peak noted in the male nomogram is likely due to under-sampling in the younger participants.”

2) Is it possible to quantify the improvement when adding the genetic information to the nomograms? See also point 6) below.

We have performed ANOVA tests to assess if the impact of genetic adjustment to the nomograms is statistically significant. We have added details of this in the methods and Results section in addition to a supplementary table with further details.

Lines 506-509 (Genetic Adjustment methods section): “To evaluate the impact of genetic adjustment, we perform a series of ANOVA tests across adjusted nomograms. E.g., we performed an ANOVA test of the HV percentiles of the top 30% UKB samples in the unadjusted then the adjusted nomograms. We did the same for bottom 30% and for men and women.”

Lines 218-220 (Genetic Adjustment Results section): “An ANOVA test of the percentiles produced with the adjusted vs unadjusted nomograms revealed that the groups were significantly different to each other (F>19; P<8.04e-6; Table 2).”

3) Line 152: "… and scan date were regressed out of the TVs" How? Is it reasonable to assume that the scanner drift is linear (the Github scripts seem to suggest this is what was modelled) but this also suggests e.g. no scanner updates, hardware changes, and so on? Was there also a correction for the different scanners that may have been used (as far as I am aware, UKB has several imaging sites).

Thank you for bringing this up. Yes, UKB uses different sites. However, they aimed to minimize site effects by using identical hardware, software etc. In a recent extensive work Alfaro-Almagro et al^9^ analyzed different sets and modeling approaches for confounds that capture all these variations. In their ‘simple’ set they included scan date. This resource was not available at the start of the study. Thus, we followed the experiments outlined in Nobis et. al (2019)^2^ as one of the main aims was to compare our performance to those models.

4) Line 220: What is the rationale for splitting high-versus low PGS at 30%? What happens at the other thresholds? Why is there a different choice for ADNI?

The 30% provides a good trade-off between sample size and performance. Too small of a percentage, 10% for example would give us closer to 140 mm3 separation in either direction but the sample sizes would be ~1.5k per strata. 50% would give us opposite results. We have added a supplementary figure (Figure 2 —figure supplement 4) that shows what the GPR models look like with different cut-offs. The ADNI subjects did not need the same considerations since they were used for testing and not to build any of the normative models; hence a split in half was sufficient to test against the appropriate nomograms.

5) Line 239: The dropout number for HV in ADNI is pretty large and probably non-random. Please comment.

This drop out is the result of looking at the quality scores provided by ADNI. The columns (RHIPQC and LHIPQC) in ADNIMERGE specifies a Pass/Fail/NA Quality Score for the hippocampal volumes coming out of Free-Surfer. We use all HV that are not scored as “Fail”. We performed a Fisher’s Test between dropout and diagnosis and found no significant correlation (P-value = 0.1776).

6) What is the meaning of {plus minus}30% in statements like "cognitively normal (CN) participants (n = 225) had a mean bilateral HV percentile of 41% ({plus minus}30%)"? Is it standard deviation/standard error? These errors seem rather large, so that leads me to believe that the e.g. 4% drop could be too small to be meaningful.

The plus minus here refers to standard deviation. We have edited the lines in question (Lines 245-253) to specify this. The 4% is indeed small (it is 6.5% after NeuroCombat adjustment), but it is driven down by the AD group having a 0.5% reduction; whereases with the CN and MCI groups we have 9% and 10% decrease in intra-group variance.

7) Discussion, first paragraph: "Therefore, accounting for … " This statement seems to contradict the results. Maybe this discussion is better placed elsewhere.

We have revised the statement it now reads:

Lines 330-334: “Therefore, in theory, accounting for the genetics of healthy variation in HV should enable earlier detection of AD-related HV decline aging individuals. Conversely, stratifying by *APOE*-e4 status when creating HV nomograms charts the different HV trajectories among *APOE* genotypes, however, at the same time masks the pathological decline and thus will theoretically decrease the sensitivity to HV decline.”

8) Discussion, second paragraph / Figure 3 / Supplementary FiguresS1/S2 / Supplementary Table S1.The (supplementary) Figures are very misleading if you compare these with supplementary Table 1: from Table 1 I conclude that every threshold predicts HV about equally well, but the figure suggests otherwise if you do not pay attention to the cut-off in the y-axis. The paragraph in the discussion that describes the so-called bimodal distribution supports this (false) idea and should be removed.

We agree that the overall predictive performance is quite similar across PGS thresholds. We have tried to make the y-axis in the supplementary figure clearer. We have adjusted Figure 3 so that the y-axis’s minimum value corresponds to the R^2^ of the model not including any PGS. And we have adjusted the phrasing do de-emphasize the bimodal nature:

Lines 180-183: “In all tested settings, the explained variance (R^2^) by the PGS across p-value threshold was similar: with one peak at the 1E-7 threshold (capturing few but very significant SNPs), a higher peak at the 0.75 threshold (capturing many SNPs with mostly small effect sizes)”.

Lines 338-339: “Furthermore, similar R^2^ values across PGS thresholds (±0.05 R^2^) with two peaks at thresholds of 1E-7 and 0.75. “

9) Discussion, Line 423: "Therefore, other brain regions with higher heritability like the cerebellum or whole brain volume may show more sensitivity on nomograms." I am somewhat confused about this sentence. Do the authors mean to imply that structures with higher heritability might benefit more from stratifying on PGS? This would only be the case if not only heritability but also SNP heritability should be higher (and the latter also depends on the genetic architecture and discovery sample size).

Thank you for raising this point. Yes, all things being equal such as sample size of the discovery GWAS, we expect brain regions with higher SNP heritability to benefit more from the PGS adjustment. We have clarified that we refer to SNP heritability in the sentence.

10) Discussion, final sentence, the brain age gap has not been mentioned in the paper up to this point. While potentially relevant, it is strange to introduce it in the final sentence.

We have changed the sentence and removed the link to brain age gap.

11) Ethics: I would have expected some statements about the use of human data from UKB and ADNI in this paragraph.

We have added and ethics statement.

12) Supplementary Figure S5: there are people that seem to switch diagnosis from AD back to MCI, this cannot be right?

It does not happen often, but these ‘reversions’ are observed in ADNI. There are subjects in ADNI reported to have gone from and to MCI/CN/AD. See RIDs: 4979, 4899, 4845, 4741, 4706, 4641, 4499, 4381, 4430, 4072, 4005.

13) Throughout the paper there are statements like "Importantly, this magnitude corresponds to ~3 years' worth of HV loss during normal aging." This suggests a constant loss over the lifespan (i.e. a linear pattern with age, but the data shows a different pattern. Please rephrase.

We are comparing the average loss across the whole lifespan to the loss found in papers like C. R. Jack, Jr. et al. (2009)^10^ and Scahill et al. (2003)^11^, where annual loss across lifespan is reported. This is how we make the claim that the loss corresponds to x amount of years of normal aging loss. To make the statements more accurate, we have taken as an example someone who is 65 years old, and we have changed all the places in the paper where the 3 years are mentioned to reflect this.

14) The (Supplementary) Figures could use a little bit more attention:- A little bit more information on what is shown in the figures is needed to be able to assess what is displayed; e.g. add abbreviations to the captions, there are no units for some of the axes. None of the nomogram figures have labels for percentile lines, which is essential. Figure S1&S2 please explain the percentile figures.

We improved the presentation of all supplementary Figures. We addressed the specific issues mentioned here and more.

References:

1. Fortin JP, Cullen N, Sheline YI, et al. Harmonization of cortical thickness measurements across scanners and sites. *NeuroImage*. 2018;167:104-120. doi:10.1016/j.neuroimage.2017.11.024

2. Nobis L, Manohar SG, Smith SM, et al. Hippocampal volume across age: Nomograms derived from over 19,700 people in UK Biobank. *NeuroImage: Clinical*. 2019;23doi:10.1016/j.nicl.2019.101904

3. Gramacy RB. LaGP: Large-scale spatial modeling via local approximate Gaussian processes in R. *Journal of Statistical Software*. 2016;72(1):1-46. doi:10.18637/jss.v072.i01

4. Mendez MF. Early-Onset Alzheimer Disease. W.B. Saunders; 2017. p. 263-281.

5. Rabinovici GD. Late-onset Alzheimer disease. Lippincott Williams and Wilkins; 2019. p. 14-33.

6. Bethlehem RAI, Seidlitz J, Romero-Garcia R, Trakoshis S, Dumas G, Lombardo MV. A normative modelling approach reveals age-atypical cortical thickness in a subgroup of males with autism spectrum disorder. *Communications Biology*. 2020-12-01 2020;3(1)doi:10.1038/s42003-020-01212-9

7. Veldsman M, Nobis L, Alfaro-Almagro F, Manohar S, Husain M. The human hippocampus and its subfield volumes across age, sex and APOE e4 status. *Brain Communications*. 2021;3(1)doi:10.1093/braincomms/fcaa219

8. Ching C, Abaryan Z, Santhalingam V, et al. Sex differences in subcortical aging: A nomogram study of age, sex, and apoe (N = 9,414). *Alzheimer's & Dementia*. 2020;16(S4):e045774-e045774. doi:10.1002/alz.045774

9. Alfaro-Almagro F, McCarthy P, Afyouni S, et al. Confound modelling in UK Biobank brain imaging. *Neuroimage*. 01 01 2021;224:117002. doi:10.1016/j.neuroimage.2020.117002

10. Jack CR, Petersen RC, Xu Y, et al. Rates of hippocampal atrophy correlate with change in clinical status in aging and AD. *Neurology*. 2000;55(4):484-489. doi:10.1212/wnl.55.4.484

11. Scahill RI, Frost C, Jenkins R, Whitwell JL, Rossor MN, Fox NC. A longitudinal study of brain volume changes in normal aging using serial registered magnetic resonance imaging. *Archives of Neurology*. 2003;60(7):989-994. doi:10.1001/archneur.60.7.989

[Editors' note: further revisions were suggested prior to acceptance, as described below.]

The manuscript has been improved but there are some remaining issues that need to be addressed, as outlined below:1. Regarding tests for Gaussianity in the UKB samples. We recommend in Figure 4 —figure supplement 3 that the interpretation of the Shapiro-Wilks test is clarified. That is, state explicitly that a given distribution is designated as non-Gaussian if the SW test yields p below some threshold. Also, we believe it is the "Shapiro-Wilk" or "Shapiro-Wilks" test, not "Shapiro-Wilkens".

Thank you for pointing out the miss-spelling. We have fixed it and edited the figure description so that it now reads:

“Figure 4 —figure supplement 3: Training Data Ridge Plots. Histograms of bilateral HV across the different subsets of the datasets. Samples are grouped in bins of 5 years. N is the number of samples in each set and p is the p-value from a Shapiro-Wilks test of normality. Typically, this test would indicate a non-gaussian distribution with a p-value lower than 0.05 (0.001 corrected for 48 multiple tests in this case).”

2. Throughout, the authors use the term "PGS score" which would be written in full as "polygenic score score". We appreciate the awkwardness that sometimes comes with acronyms, but suggest sticking to either "PGSs" or "PG scores".

Thank you for the appreciation, we have edited the manuscript to stick with ‘PGSs’ throughout.

3. It might be worthwhile adding some discussion regarding Reviewer 1's comments about the potential benefits of directly incorporating PGSs in normative modelling, alongside the challenges that the authors raise in their response letter.

We have added the following lines in the discussion:

Lines 380-384: “While this study has shown the significant impact of PGSs on HV nomograms, we have identified areas for improvement. Integrating the PGSs into the GP models would remove the need for stratification and allow for more adjustment precision, however, PGSs are prone to ‘site’ effects depending on the method and quality of genotyping and imputation. Consequently, using the ‘raw’ PGSs in predictive models presents its own challenges.”

4. It might help readers less familiar with sliding window techniques to be even more explicit about the reason why smoothing restricts the age range. The authors state this but do not note that this is due to "edge effects", in which smoothed sliding window curves become highly sensitive to noisy data at the limits of data ranges.

We have added some clarification in the results and methods sections.

Lines 145-147 (Results section): “This is primarily because the SWA is a non-model-based approach that requires smoothing to avoid edge effects, and a gaussian smoothing window of width 20 was used”

Lines 458-460 (methods section): “. The quantiles were then smoothed with a gaussian kernel of width 20. The smoothing was needed because towards the ends of the data, the sliding windows approach becomes sensitive to noise.”

5. The new results in Figure 5 might be better visualised as violin or raincloud plots. However, we do appreciate that Reviewer 2, who requested this information, did also suggest that boxplots would suffice.

Thank you for the suggestion. We agree that the violin plots provide a clearer sense of the data distribution and have remade the figure with split violin plots and included them in the manuscript.

6. Please consider dampening the conclusions ever so slightly. NeuroCombat generally does an excellent job at removing some site related variation, but does not remove the tenacious issue of site effect altogether.

We have added a sentence towards the end of the discussion when mentioning the limitations of the study.

Lines 395-397: “Finally, while NeuroCombat has been shown to remove most site effects, some may remain and still need to be adjusted for.”